# Perspectives of Cell Sensitivity/Resistance Assay in Soft Tissue Sarcomas Chemotherapy

**DOI:** 10.3390/ijms241512292

**Published:** 2023-07-31

**Authors:** Timur I. Fetisov, Sofya A. Khazanova, Polina A. Shtompel, Ekaterina S. Trapeznikova, Victoria Y. Zinovieva, Valeria I. Marshall, Anastasia A. Lovenger, Dmitriy V. Rogozhin, Tararykova A. Anastasia, Beniamin Yu. Bokhyan, Gennady A. Belitsky, Marianna G. Yakubovskaya, Kirill I. Kirsanov

**Affiliations:** 1N.N. Blokhin National Medical Research Center of Oncology, 115478 Moscow, Russia; timkatryam@yandex.ru (T.I.F.); anastasiatararykova@gmail.com (T.A.A.);; 2Faculty of Medicine, Sechenov University, 119991 Moscow, Russia; trapeznikova723@gmail.com; 3Institute of Medicine, RUDN University, 117198 Moscow, Russia

**Keywords:** soft tissue sarcoma, undifferentiated pleomorphic sarcoma, synovial sarcoma, chemoresistance, doxorubicin, ifosfamide, gemcitabine, docetaxel

## Abstract

Treatment of highly malignant soft tissue sarcomas (STSs) requires multicomponent therapy including surgery, radiotherapy, and chemotherapy. Despite the advancements in targeted cancer therapies, cytostatic drug combinations remain the gold standard for STS chemotherapy. The lack of algorithms for personalized selection of STS chemotherapy leads to unhelpful treatment of chemoresistant tumors, causing severe side effects in patients. The goal of our study is to assess the applicability of in vitro chemosensitivity/resistance assays (CSRAs) in predicting STS chemoresistance. Primary cell cultures were obtained from 148 surgery samples using enzymatic and mechanical disaggregation. CSRA was performed using resazurin-based metabolic activity measurement in cells cultured with doxorubicin, ifosfamide, their combination and docetaxel, gemcitabine, and also their combination for 7 days. Both the clinical data of patients and the CSRA results demonstrated a higher resistance of some cancer histotypes to specific drugs and their combinations. The correlation between the CSRA results for doxorubicin and ifosfamide and clinical responses to the combination chemotherapy with these drugs was demonstrated via Spearman rank order correlation. Statistically significant differences in recurrence-free survival were also shown for the groups of patients formed, according to the CSRA results. Thus, CSRAs may help both practicing physicians to avoid harmful and useless treatment, and researchers to study new resistance markers and to develop new STS drugs.

## 1. Introduction

Chemotherapy (ChT) is a critical component of therapy for stage 2–3 soft tissue sarcomas (STSs) [1]. However, tumor responses to ChT may vary significantly among patients due to the high diversity of STSs, which becomes a major problem in the management of this type of cancer [2]. The most common ChT courses use combinations of doxorubicin (Dox) and ifosfamide (IFO), as well as docetaxel (Doc) and gemcitabine (Gem) [3]. It is particularly noteworthy that a positive response to therapy is observed only in 20–25% cases, tumor stabilization is observed in about 50% cases, and no effect on tumor growth is observed in 25–30% [3,4]. It turns out that over 25% of STS patients receive treatment with highly toxic drugs, leading to harmful effects. It makes the development of personalized methods predicting the tumor response to ChT increasingly relevant.

The research efforts to predict STS response to ChT can be divided into two main approaches, which give positive results for other types of cancer. The first one is based on the use of molecular biological markers for ChT-resistance, and the second one consists of testing the effects of drugs in cells isolated from the tumors [1,2,3,4]. Currently, there are no clinically used molecular markers that predict responses to therapy and survival in STSs. The data obtained from some STS cell lines [5] are not in full agreement with the results of tumor analysis of STS patients [6]. Although the mutation rate is low in STS [7], a number of studies have revealed a relationship between the development of resistance to Dox and the presence of mutations in the apoptosis activation pathway [8]. It has been found that in ChT-resistant STS samples, the expression of genes involved in signaling pathways associated with cell adhesion, migration, proliferation, cytotaxis, and phagocytosis, as well as signaling cascades PI3K-Akt, NF-ĸB, MAPK, and Jak –STAT is activated. *C1QC*, *C3AR1*, *C7*, *CFI*, and *SERPINE1* were thus selected as potential ChT resistance and survival biomarkers [9]. Lindner et al. showed that a high expression of TOP2A was associated with low overall survival, while high expression of SIRT1 was associated with high overall survival [10]; according to the authors, the presented markers may be of interest for predicting the survival of patients with STSs.

An alternative approach consisting of experimental ChT resistance testing via a Cell Sensitivity assay (CSA) was proposed by Kurbacher at al. for ovarian cancer [11]. Using a CSA, a number of studies of tumors of different histogenesis have been published, and a high prognostic value of predicting resistance to chemotherapy has been demonstrated for different cancers [12]. Volm and Efferth paid more attention to in vitro testing of the ChT-resistance than to ChT-sensitivity, as in several clinical trials, ChT-resistance was found to be predictable, with more than 90% accuracy, while drug sensitivity was predicted with significantly less accuracy (~60%). Consequently, we mainly directed our study to ChT-resistance prediction, in order to develop a future approach for safer chemotherapy by preventing useless and harmful treatment of STS patients, and we have extended the abbreviation, CSA, proposed by Kurbacher at al., to CSRA (Cell Sensitivity/Resistance Assay).

CSA was used for STS in only a few studies. M. Lehnhardt et al. studied the correlation between the CSA data to ChT drugs, tumor histotype, and malignancy grade using tumor samples of 50 STS patients [13]. In later years, two clinical cases using the studied approach were published detailing the selection of neoadjuvant ChT regimens in patients with synovial sarcoma [14] and undifferentiated pleomorphic sarcoma [15]. In our previous studies, using the CSRA test, we showed that mutations in apoptosis signaling pathways are associated with resistance to Dox [8], and it was also shown that the expression of ABC transporters in adult STSs does not play a significant role in ChT-resistance [10]. Oncomedics have proposed a similar to CSA method for ChT-resistance experimental testing in colon cancer [16] and ovarian cancer [17]. There are a number of differences between oncomedics and CSA, including different protocols for tumor sample treatment, seeding, and culturing conditions. Currently, we have performed the study to compare these methods.

Previously, at the Department of Chemical Carcinogenesis of N.N.Blokhin NMRCO, the methods ensuring enzymatic disaggregation of tumor cells for different STS types were tested, the cell culture conditions were optimized, and the drug doses determined. In the present study, using 148 STS samples, we performed CSRA for four ChT drugs (Dox, IFO, Gem, and Doc) and two drug combinations (Dox + IFO and Doc + Gem) commonly assigned as the first- and second-line of STS therapy. Simultaneous treatment of cells in primary cultures with Dox and IFO in CSRA corresponded to the most used protocol of CSA proposed by Kurbacher at al. [15], and it was found to be concordant with the protocol for STS patient treatment when Dox and ISO were used in combined chemotherapy.

At the final stage of our study, the CSRA results were compared with the clinical data on the course of the disease for each patient.

## 2. Results

The total of 148 samples from STS patients were obtained after surgery at N.N. Blokhin National Medical Research Center of Oncology. The patients were of an average age of 51 (ranging from 12 to 88) with a slight prevalence of females (78, 53%). A total of 91 (61%) cases were newly diagnosed; the prevalent sites of tumors were extremities with 108 cases (73%), among which, 81 tumors were located in lower limbs (75% of upper and lower limb tumors combined). The distribution by histotypes was as follows: pleomorphic sarcomas (PS)—41 (29%); liposarcomas (LPS)—39 (26%); synovial sarcomas (SS)—25 (17%); leiomyosarcomas (LMS)—14 (9%); other—29 (20%). Myxoid liposarcomas were the prevalent LPS type with 20 cases (51% of all liposarcomas). Grade 3 was the prevalent malignancy grade with 108 cases (73%). Summing up information concerning the samples is presented in Table 1.

Cytological analysis performed after cell isolation from all primary cultures showed an average content of tumor cells of 80%, and after a week of culturing, the average content of tumor cells increased to 95%. Resistance of tumor cells to Dox, IFO, and their combination (Dox + IFO) as well as to Gem, Doc and also their combination (Doc + Gem) was tested on STS primary cultures (Figure 1). The sensitivity/resistance index (SRI), which was calculated based on metabolic activity of tumor cells cultured to the drugs (Materials and methods), was used as an indicator of tumor cell sensitivity/resistance to the drugs and their combinations. The CSRA results with SRI higher than 250 were considered resistant to the drug, and the results of SRI lower than 250 were considered as sensitivity prediction. Results of CSRA for 148 patients arranged by SRI color scale are presented in Figure 1.

The Dox + IFO combination turned out to be more effective than IFO, Doc, and Gem (*p* = 0.03; *p* < 0.0001), and cell resistance to the Doc + Gem combination was lower than Doc, Gem, and IFO (*p* = 0.004; *p* < 0.0001; *p* = 0.0003). At the same time, no significant difference was observed between SRI to Dox and Dox + IFO. In addition, tumor cells were less resistant to Dox + IFO than to Doc + Gem (*p* = 0.0025). Among the tested cultures 38% were resistant to both combinations, and 23% were sensitive to both.

SRI values obtained for different drugs showed positive correlation (with r ranging from 0.21 to 0.81, *p* < 0.0001). In particular, SRI (Doc) and SRI (Gem) were correlated with each other and strongly correlated with SRI (Doc + Gem) (r = 0.51; r = 0.75 and r = 0.81, respectively; *p* < 0.0001). SRI (Dox) and SRI (IFO) showed a weak positive correlation with each other (r = 0.31; *p* < 0.0001) and were moderately correlated with SRI (Dox + IFO) (r = 0.68 and r = 0.76, respectively; *p* < 0.0001). In addition, SRI (Dox) was moderately correlated with SRI (Doc) and SRI (Gem), as well as SRI (Doc + Gem) (r = 0.47; r = 0.45 and r = 0.49, respectively; *p* < 0.0001); whereas SRI (IFO) was weakly correlated with SRI(Doc), SRI (Gem), and SRI(Doc + Gem) (r = 0.21; r = 0.27 and r = 0.31, respectively; *p* < 0.0001). SRI (Dox + IFO) showed moderate correlation with SRI (Doc + Gem) (r = 0.45, *p* < 0.0001).

Comparison of SRIs of patient groups formed according to age, malignancy grade, ChT history, and recurrence history did not reveal any significant difference (Table 2), while primary cell cultures obtained from females were more resistant to Dox, respectively, to the cultures obtained from males (Table 2, Figure 1B). It is noteworthy that the most significant difference in SRI between the cultures obtained from males and females was observed for PS. The dependence of SRI values on STS histological types (LPS, PS, SS, and LMS) was also analyzed. PS cells showed higher Dox resistance than SS and LPS cells (*p* = 0.015 and *p* = 0.024, respectively), while LMS were more sensitive to Gem than SS (*p* = 0.024) and more sensitive to the Doc + Gem combination than LPS (*p* = 0.027) (Table 2).

Among 148 patients included in CSRA, 66 received the Dox + IFO neoadjuvant and/or adjuvant chemotherapy regimens and underwent regular examination at the Blokhin NMRCO. We separated them into two groups depending on their response to ChT, and compared SRI in the subgroups. It was found that SRI for IFO and the Dox + IFO combination were significantly higher in the group of patients lacking any clinical response to the administered ChT (negative changes, sustained growth, or tumor development) than in the group characterized by a good clinical response to the ChT (tumor stabilization, partial response, or complete tumor regression) (*p* < 0.0001 and *p* = 0.01, respectively) (Figure 2A). In addition, in the group of patients lacking ChT response, the number of patients with SRI predicting chemoresistance exceeded the corresponding number in the group of patients with a good ChT response (Figure 2B).

Unfortunately, we were unable to put together a subgroup of patients receiving Doc + Gem therapy due to their underrepresentation.

The recurrence data were obtained for 115 patients. This group included patients, who received Dox- and IFO-based ChT, radiation therapy, as well as those who only received surgery. It was shown that the group of patients with CSRA results predicting ChT-resistance to IFO and its combination with Dox had significantly lower recurrence-free survival (RFS) than the group with SRI corresponding to ChT-sesitivity (HR = 1.892; 95% CI 1.179 to 3.037; *p* = 0.0082 and HR = 2.226; 95% CI 1.332 to 3.718; *p* = 0.0022). Furthermore, a trend for lengthier RFSs was observed in the group of patients with SRI predicting ChT sensitivity to Dox and the Doc + Gem combination (HR = 1.537; 95% CI 0.9634 to 2.452; *p* = 0.0713 and HR = 1.494; 95% CI 0.936 to 2.384; *p* = 0.0926). On top of that, statistical significance was shown using Gehan–Breslow–Wilcoxon test, which assigns higher weights to the events at earlier time points (*p* = 0.0086 and *p* = 0.048) (Figure 3).

After that, we formed a group of 77 patients, who were included in CSRA, received Dox + IFO as their neoadjuvant and/or adjuvant chemotherapy, both at the NMRCO and at the local oncological centers, and provided the data for RFS. The patients with CSRA results predicting ChT-resistance to Dox and its combination with IFO showed shorter RFSs than the group with SRI predicting ChT-sensitive to the same therapy (HR = 1.732; 95% CI 1.015 to 2.954; *p* = 0.0438 and HR = 2.355; 95% CI 1.269 to 4.371; *p* = 0.0066). Herein, even shorter RFSs were observed in patients with SRI corresponding to ChT resistance to the Doc + Gem combination (HR = 1.709; 95% CI 1.001 to 2.918; *p* = 0.0497) (Figure 4).

## 3. Discussion

STS therapy personalization is a challenge, wherein oncologists must select a therapy with the lowest likelihood of ineffectiveness with severe side/toxic effects. Nowadays, it is impossible to imagine advanced oncological practice without validated molecular biological methods allowing physicians to make decisions taking into account individual molecular markers of disease progression. However, in STS therapy, physicians typically continue to choose one of three ChT regimens, in particular, Dox monotherapy, Dox + IFO, or Doc + Gem [18,19], considering only the STS histotype and the stage of tumor progression. The study of the long-term ChT results showed that the group of patients who received Dox + IFO therapy based on histological analysis was not superior in RFS and overall survival compared to the group of patients who were assigned the same therapy without the mentioned analysis [19]. This can possibly be explained by high internal diversity of STSs and the presence of resistant tumor cells, which require further development of personalized approaches to ChT decision-making.

The correlation between CSRA results predicting resistance to specific drugs and time to recurrence was demonstrated in our study, which emphasizes the applicability of the studied tests in the specified disease types. It was found that the high resistance to Dox, Dox + IFO, and Doc + Gem in CSRA agrees quite well with the clinical data on the efficacy of the selected drugs in STS [18,20,21]. Some dependence of SRIs on STS histotypes was also observed. Dox and IFO demonstrated the highest cytotoxic effect on SS and LPS in CSRA, while the lower resistance to Gem and its combination with Doc was observed in LMS cells, which is also confirmed by the actual clinical data showing SS and LPS as the most susceptible to ChT [22]. The difference between tumor cell resistance to Dox observed for males and females is noteworthy as well. Although not directly confirmed by the clinical data from STS patients [23,24], these effects may be associated with protective effect of signaling pathway proteins of gonadal hormones. It has been shown that administration of antiestrogen drugs intensifies the anti-tumor activity of cisplatin [25], and progesterone activates the expression of a key multidrug resistance gene—BCRP1 [26].

The presence of resistant tumor cells is among the main causes of recurrences, and tumor cells with metastatic phenotype demonstrate low sensitivity to ChT drugs [27]. The sample of patients used in our study included the subgroup who received ChT, radiation therapy or surgery alone, and it was used to demonstrate that the resistance to IFO, Dox + IFO, and Doc + Gem predicted by CSRA significantly correlated with the RFS. Using the subgroup of patients who received Dox + IFO therapy, we demonstrated that in vitro resistance to Dox + IFO correlated with the patient response to ChT. It was also found that in the group with the resistance to Dox and its combination with IFO, as revealed by CSRA, there was significant reduction of the RFS. A similar peculiarity was observed in patients with resistance to Doc + Gem predicted by CSRA, which most likely indicates the presence of the mechanisms developing cross resistance to both drug combinations. Tumor cell resistance to therapeutic agents should be associated with genetic background defining the difference in sensitivity to various agents (e.g., the presence of gene isoforms in the genome whose proteins slow down/accelerate drug metabolism in the organism), cell-mediated factors influencing drug transit to the cells and from tumor cells, activity of the systems of DNA repair, cell death, as well as factors mediated by the interactions of tumor cells with their microenvironment (changes in the extracellular matrix, activation of non-tumor cells) [28]. Tumor cell resistance to the drugs identified via the CSRA results mainly from the tumor cell genetics and cell-mediated factors. At the same time, this test minimizes the influence of a number of factors on the development of ChT resistance, in particular, factors mediated by the interaction of cancer cells with the microenvironment, as well as stem and hypoxic phenotypes, which may impose some limitations on the prognostic significance of CSRA-revealed sensitivity [16].

Summing up, CSRAs can become an effective tool for the prediction of STS tumor resistance to anticancer drugs and their combinations, providing information for the correction of ongoing treatment and exclusion of a drug from the combination in the presence of resistance to it. Choosing the type and duration of STS chemotherapy remains an unresolved problem [29,30] in which the CSRA approach may be useful in providing an informed choice between the drug combinations. In addition, this test can be used both in a search and selection of prognostic biomarkers and in a study of resistance mechanisms in STS cells. We have already shown the effectiveness of the CSRA approach in the analysis of STS multidrug resistance caused by ATP-binding cassette transporters and apoptosis inactivation [6,8]. Thus, CSRA application in the study and management of STSs will contribute to a more rapid progress of knowledge in this field.

## 4. Materials and Methods

### 4.1. Tumor Samples

The study was approved by the Ethics Committee of the N.N. Blokhin National Medical Research Center of Oncology (Moscow, Russia). Written informed consent was given by all the patients. All procedures were performed in compliance with the WMA International Code of Medical Ethics. From November 2018 to March 2022, 147 STS samples were obtained from the Research Institute of Clinical Oncology of the N.N. Blokhin National Medical Research Center of Oncology. The studied tumor samples were taken from patients diagnosed with STS, with the diagnoses were confirmed with histology.

### 4.2. Chemotheraputic Drugs

The drugs tested in the study were as follows: Dox (Doxorubicin, RONC, Moscow, Russia), Doc (docetaxel, NATIVA, Moscow, Russia), and Gem (Gemcitabin, BIOCAD, Moscow, Russia). IFO is a prodrug requiring in vivo liver activation, which is why an active metabolite 4-hydroxy-ifosfamide (4-OH-Ifo, NIOMECH, Bielefeld, Germany) was used in the study.

### 4.3. Chemosensitivity Measurement

CSRA was performed as a routine procedure immediately following surgery. Tumor samples were fragmented (to the size of 1 mm^3^) and then dissociated via incubation in a 5–10 mL of sterile collagenase solution for 2–3 h at 37 °C. Single-cell suspension was prepared using a shaker. After achieving a concentration of 1–2 × 10^5^ cells/mL, 100 μL of cell suspension was added to each well of a 96-well microplate. The tested ChT drugs were added in six concentrations as follows: 6.25%, 12.5%, 25%, 50%, 100%, and 200% of standard TDC, as suggested by Andriotti et al. [31] (Table 3).

The plates were incubated for 5–6 days under standard conditions (temperature of 37 °C, humidity of 95%, 5% CO_2_). Cell viability was analyzed based on metabolic activity using resazurin, as described earlier [32]. Sensitivity/resistance index (SRI) for each tested drug was calculated as SRI = 600—sum of % TGI at 200, 100, 50, 25, 12.5, 6.25 TDC, where TGI is the tumor growth inhibition percentage and TDC is the standard testing drug concentration (% from peak plasma concentration of the tested drug). Thus, SRI = 600 indicates a complete cell viability in presence of the drug (lack of drug sensitivity), whereas SRI = 0 indicates complete cell death in presence of the tested agent (maximum sensitivity to the drug) [31]. To divide the samples into sensitive and resistant, we used a threshold value of SRI = 250 used in similar studies [33]. The samples with SRI > 250 were considered resistant to the drug, and the samples with SRI ≤ 250 were considered sensitive.

### 4.4. Cytological Analysis

The percentage of tumor cells was measured for all samples tested for chemosensitivity right after the isolation and after a week of cultivation. For this purpose, thin-layer slides were prepared using Thermo Shandon Cytospin 3 (Marshall Scientific, Hampton, NH, USA). Morphological analysis of Leishman stained preparations was carried out using Nikon Eclipse Ci-S microscope (Nikon Corporation, Tokyo, Japan) at 500× magnification in three FOVs. The percentage of malignant cells was calculated using the conventional cytological criteria, including cell shape, cytoarchitecture, and parameters associated with genome instability [34].

### 4.5. Statistical Analysis

Statistical processing of the results was performed using GraphPad Prism 8.4.3 (GraphPad Software, San Diego, CA, USA). Quantitative assessment of correlation between the studied parameters was carried out using the Spearman correlation coefficient, and Kruskal–Wallis test was used to compare the studied groups. RFS lengths in resistant and sensitive groups were compared using Gehan-Breslow-Wilcoxon test and logrank test.

## Figures and Tables

**Figure 1 ijms-24-12292-f001:**
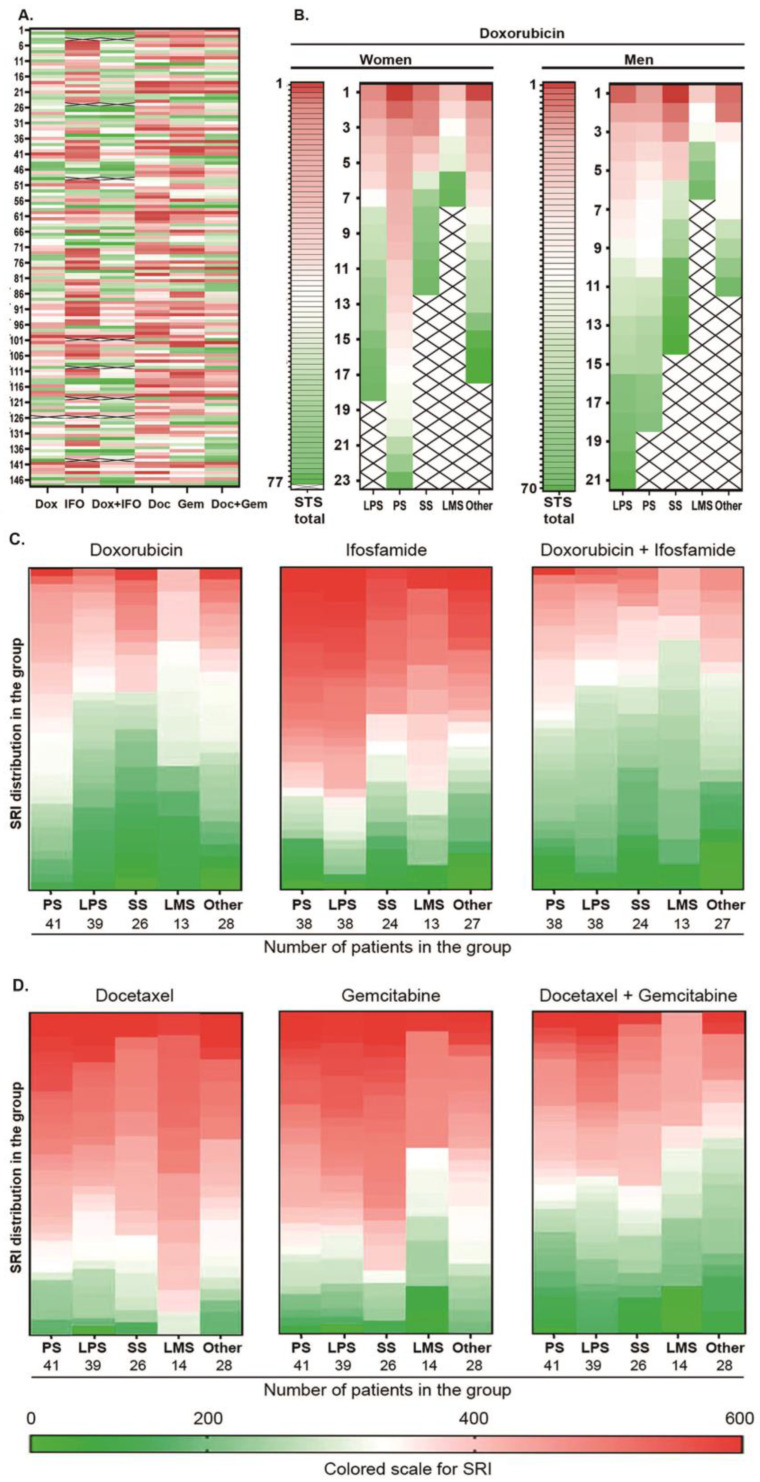
Results of CSRA presented for 148 patients by SRI corresponding to color scale; (**A**) integral hitmap of SRI of the primary cell cultures obtained from 147 STS patients for Dox, Ifo, Dox, and Ifo combination, Doc, Gem and Doc, and Gem combination; (**B**) hitmap of SRI presented by color scale of the primary cell cultures obtained from 78 female STS patients for all the drugs and combinations arranged by SRI for Dox; (**C**) hitmap of SRI presented by color scale of the primary cell cultures obtained from 69 male STS patients for all the drugs and combinations arranged by SRI for Dox; (**D**) distribution of SRI for Dox, Ifo, Dox, and Ifo combination, Doc, Gem, and Doc and Gem combination arranged by SRI by color scale in the groups of STS patients with different histologic types of tumors.

**Figure 2 ijms-24-12292-f002:**
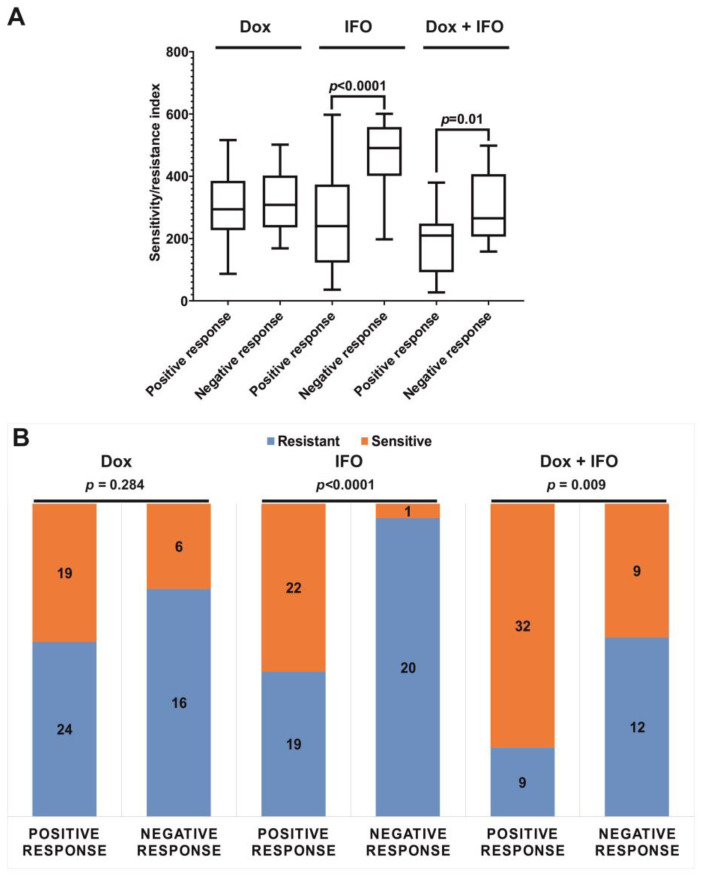
Comparison of the groups of patients who responded positively and negatively to chemotherapy by (**A**) the distribution of SRI with an indication of the total range of values, Q2-Q3 quartiles, and median values; (**B**) the numbers of resistant and sensitive primary cell cultures in CSRA.

**Figure 3 ijms-24-12292-f003:**
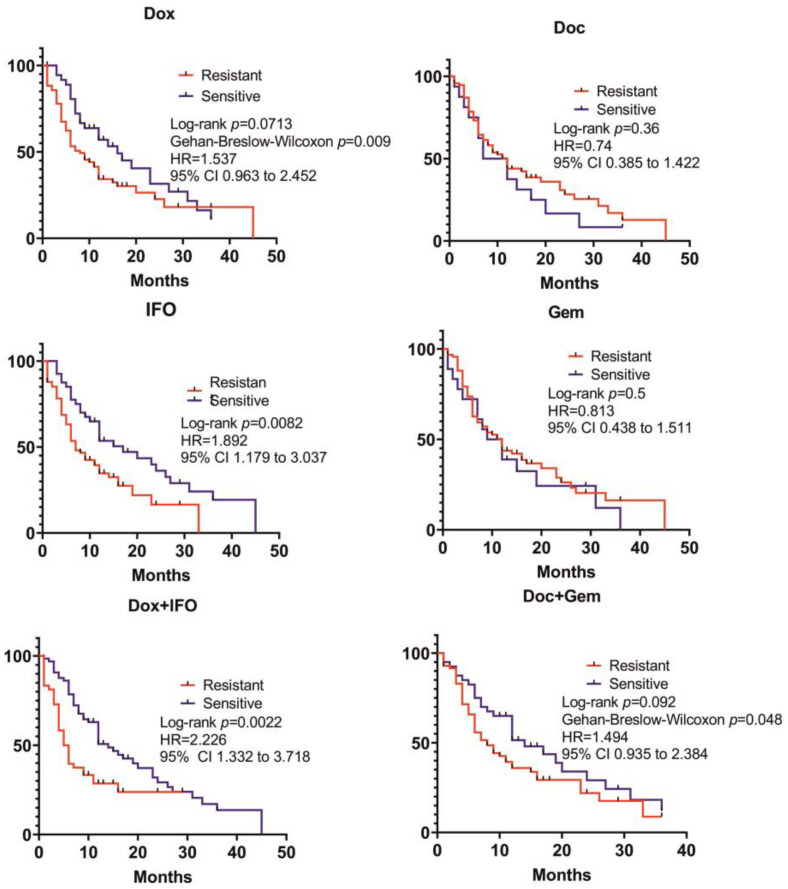
Comparison of RFS in the groups of patients with SRI corresponding to ChT-resistance or ChT-sensitivity.

**Figure 4 ijms-24-12292-f004:**
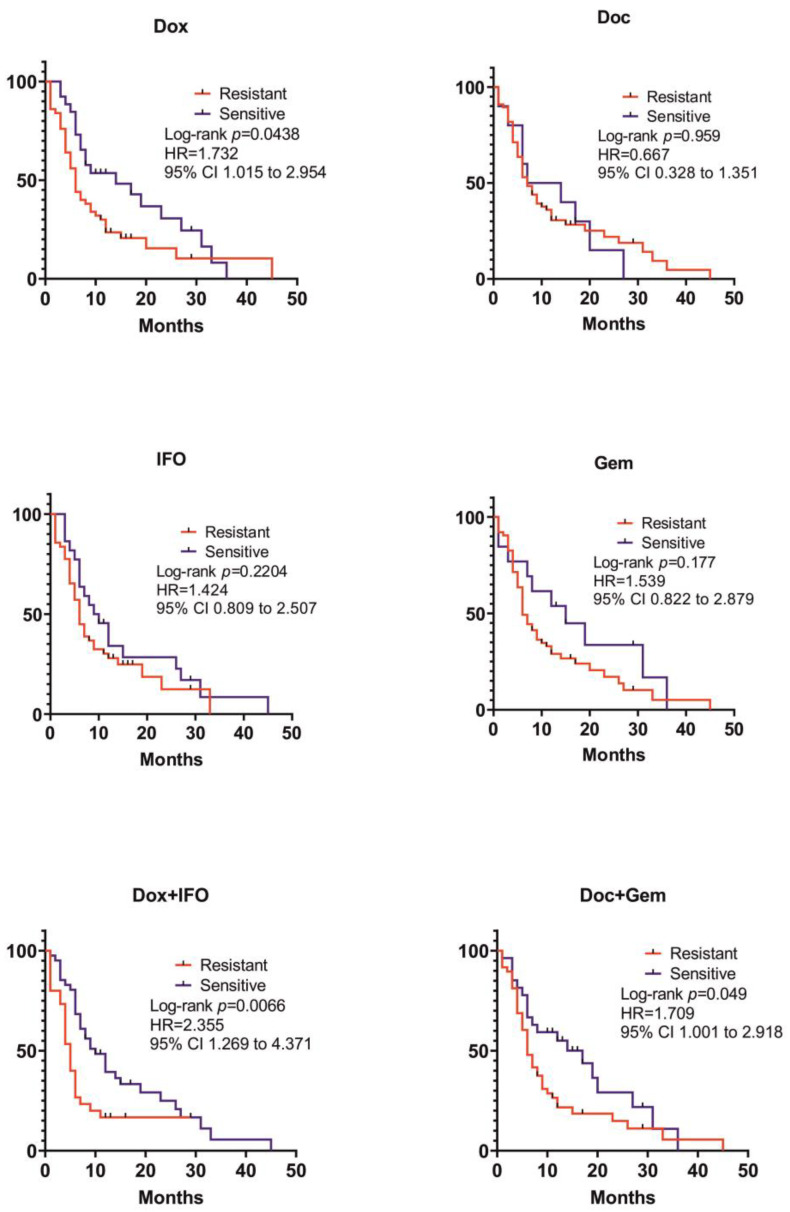
Comparison of recurrence-free survival in resistant and sensitive patients treated Dox and Ifo combination.

**Table 1 ijms-24-12292-t001:** Patient sample description.

Parameters	Quantity (%)
**Sex**	
F	78 (53%)
M	70 (47%)
**Age**	Average age: 51
<40	34 (23%)
>40	114 (77%)
**Malignancy grade**	
G1–2	40 (27%)
G3	108 (73%)
**Primary site**	
Extremities	108 (73%)
Trunk	35 (24%)
Head	5 (3%)
Newly diagnosed	91 (61%)
Recurrence	57 (39%)
**Previous ChT**	
No	84 (57%)
Yes	62 (42%)
N/A	2 (1%)
**Histology**	
Pleomorphic sarcoma (PS)	41 (28%)
Liposarcoma (LPS)	39 (26%)
Synovial sarcoma (SS)	25 (17%)
Leiomyosarcoma (LMS)	14 (9%)
Other	29 (20%)

**Table 2 ijms-24-12292-t002:** Comparison of SRI index of the patient groups formed according to the clinical characteristics, Kruskal–Wallis test.

		Drug Tested in CSRA					
Compared Groups	n	Dox	IFO	Dox + IFO	Doc	Gem	Doc + Gem
Age <40 vs. >40	148	ND *p* = 0.75	ND *p* = 0.12	ND *p* = 0.24	ND *p* = 0.41	ND *p* = 0.13	ND *p* = 0.35
Sex F vs. M	148	SD (higher resistance of STS in the female group) *p* = 0.035	ND *p* = 0.074	ND *p* = 0.076	ND *p* = 0.861	ND *p* = 0.989	ND *p* = 0.779
Malignancy grade G1–G2 vs. G3	148	ND *p* = 0.514	ND *p* = 0.277	ND *p* = 0.578	ND *p* = 0.362	ND *p* = 0.961	ND *p* = 0.359
Newly diagnosed vs. Recurrent (with no ChT)	148	ND *p* = 0.939	ND *p* = 0.5	ND *p* = 0.306	ND *p* = 0.244	ND *p* = 0.229	ND *p* = 0.25
Newly diagnosed vs. Recurrent (with ChT)	62	ND *p* = 0.28	ND *p* = 0.443	ND *p* = 0.857	ND *p* = 0.237	ND *p* = 0.859	ND *p* = 0.885
Histology Liposarcoma (LPS) Pleomorphic sarcoma (PS) Synovial sarcoma (SS) Leiomyosarcoma (LMS)		SD (higher resistance of pleomorphic sarcomas) *p* = 0.015	Lower resistance of synovial sarcomas *p* = 0.053	ND *p* = 0.861	ND *p* = 0.556	SD (lower resistance of leiomyosarcomas) *p* = 0.024	SD (low resistance of leiomyosarcomas) *p* = 0.027

ND—no difference, SD—significant difference, *p*-value—the probability under the assumption of no effect or no difference.

**Table 3 ijms-24-12292-t003:** Tested drugs and their respective 100% TDCs used in in vitro testing.

Drug/Combination	100% TDC (mg/mL)
Dox	3.0
Ifo	3.0
Dox + IFO	3.0 + 3.0
Doc	11.3
Gem	25.0
Doc + Gem	11.3 + 25.0

## Data Availability

Not applicable.

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
