# Peer review of "Perspectives of Cell Sensitivity/Resistance Assay in Soft Tissue Sarcomas Chemotherapy"

_ijms, 2023, doi:10.3390/ijms241512292_

Round 1

Reviewer 1 Report

The objective of the authors was to assess the applicability of in vitro chemo sensitivity/resistance assays (CSRA) in predicting soft tissue sarcomas (STS) response to chemotherapeutic drugs. The authors have used isolated cells from 147 soft tissue sarcoma patient samples for this study. The authors tested four common chemotherapeutic agents used in the treatment of STS; namely doxorubicin, ifosfamide, docetaxel, gemcitabine, and their combinations (Dox+IFO or Doc+Gem) on each of primary culture established from 147 patient samples. The authors have shown the tested primary cultures are more resistant to Doc and Gem than Dox and IFO. The authors have also shown that the primary tumor cells are more sensitive to Dox+IFO combination compared to Doc+Gem combination. The authors have also compared the results from CSRA with clinical data. It is very interesting data and the manuscript could be considered for publication.

Please proofread the manuscript for syntax errors

Author Response

Dear Peer Reviewer,

We are very grateful for your careful reading of our manuscript and your comments on the significance of our study. We corrected typing errors and carefully checked English. Thank you for your review!

Reviewer 2 Report

The manuscript by Fetisov et al aims at assessing the applicability of in vitro chemosensitivity/resistance assays in predicting chemotherapy response of soft tissue sarcomas. Although interesting, several comments need to be addressed

Major comments I am still confused about Table 2. The standard deviation seems huge and thus having that information in the first row is pretty much pointless. Would it be possible to make it as a bar graph or a heatmap in order to show the differences in metabolic activity to the different drugs based on the different histologies? The higher resistance of sarcomas in the female group is very interesting, try to show it also based on histology Aren't panel A and B of figure 1 giving the same information? What is the point? Figure 1, 2 and 3: Resistant, not resistive; Log-rank, not long-rank Figure 3 legend: treated by AI?     Minor comments I would move the line 57-61 to the discussion Can you add a reference to line 64? If not, move to the discussion Use LMS as abbreviation of leiomyosarcoma, instead of LeS Move line 247-248 to line 88 in order to make clearer what the levels of SRI mean (i.e. above or below 250) Use abbreviations for histologies in lines 112-113

Author Response

Dear Peer Reviewer,

We are very grateful for your careful reading of our manuscript, its thorough analysis and useful comments. We appreciate your rational pieces of advice very much and did our best to improve our manuscript following your recommendations. Please find our responses on all your comments point by point.

Comment 1: I am still confused about Table 2. The standard deviation seems huge and thus having that information in the first row is pretty much pointless. Would it be possible to make it as a bar graph or a heatmap in order to show the differences in metabolic activity to the different drugs based on the different histologies? The higher resistance of sarcomas in the female group is very interesting, try to show it also based on histology.

Response 1: Following your advice we removed first line from the Table 2 and added the new Figure 1: Results of CSRA presented for 148 patients by SRI correspondingly to colored scale; (A) Integral hitmap of SRI of the primary cell cultures obtained from 148 STS patients for Doxorubicin, Ifosphamide, Doxorubicin and Ifosphamide combination, Docetaxel, Gemcitabine and Docetaxel and Gemcitabine combination; (B) Hitmap of SRI to Doxorubicin presented by colored scale of the primary cell cultures obtained from 78 female and 70 male STS patients; (C) Distribution of SRI for Doxorubicin, Ifosphamide, Doxorubicin and Ifosphamide combination, Docetaxel, Gemcitabine and Docetaxel and Gemcitabine combination ranged by SRI by colored scale in the groups of STS patients with different histologic types of tumors.

Comment 2: Aren't panel A and B of figure 1 giving the same information? What is the point?

Response 2: For the groups of patients, who responded positively and negatively to chemotherapy, panel A shows the distribution of SRI with an indication of the total range of values, Q2-Q3 quartiles and median values, while panel B the numbers of resistant and sensitive primary cell cultures in CSRA. We have corrected the legend of the Figure (now it is the Figure 2).

Comment 3:  Figure 1, 2 and 3: Resistant, not resistive; Log-rank, not long-rank Figure 3 legend: treated by AI?

Response 3: Mistakes have been corrected.

Comment 4:  Can you add a reference to line 64? If not, move to the discussion.

Response 4: The references have been added.

Comment 5: Use LMS as abbreviation of leiomyosarcoma, instead of LeS.

Response 5: The abbreviation has been corrected.

Comment 6: Move line 247-248 to line 88 in order to make clearer what the levels of SRI mean (i.e. above or below 250)

Response 6: It has been fixed.

Comment 7: Use abbreviations for histologies in lines 112-113

Response 7: It has been fixed

Reviewer 3 Report

In this manuscript, the authors compare clinical response from patient to sensitivity/resistance of tumor cells adapted in vitro and treated to conventional chemotherapy. The goal could be to used these in vitro information for predicting tumor response and to propose a personalized medicine. If the approach is interesting, especially for hospital where access to genetic and transcriptomic screening are difficult, the analysis remain global and statistical and this manuscript lacks too many point to discussed to be acceptable at this step.

The authors underlined the fact that development of personalized methods predicting the tumor’s response are needed for optimizing treatment by chemotherapy. However, in Introduction, they completely omitted to mention all genomic and transcriptomic investigations. Now NGS and kits for detecting mutations are usually used for profiling patient. As an example, for some cancers, detection of KRas mutations is now used for defining treatment of patients. These methods will be far more standardizable than in vitro tumor cell culture. Introduction should be enriched by presenting the other approaches.

It is also important to notice that the approach the authors propose is already commercialized by Oncomedics. That could be mentioned and discussed if the techniques proposed is not strictly identical.

Authors completely omitted to discuss all the limitations of the in vitro tumor cells. Many CROs try to transfer primary tumor cell for establishing PDX or for performing in vitro experiments. They all observed that in vitro adaptation is not uniform : some adaptation are easy but with some other tumor cells do not grow, and it is difficult to not consider a case by case situation. May be that this predictive approach is more appropriate for certain tumors but this question was even never evoked. These points have to be discussed and some in vitro curves have to be presented in a novel figure (or at least as supplementary data).

The question of the in vitro adaptation is also important because stem cell-like cells usually do not grow in vitro … but they have a major role in clinical resistance. This bias is completely omitted.

Sensitivity of drugs in vitro are mentioned but their mode of action not. It is well known that depending on the mode of action, optimal drug concentration and incubation time are not the same. Therefore, the meaning of global and statistical analysis  without any consideration of this point is questionable.

It is extremely difficult to compare combination of drugs and in clinic and in vitro. Chronology and doses of clinical treatment are far different than those of coincubation in vitro, especially with drugs that induce apoptosis after a cell cycle blockage. It essential to well explain the rational for establishing the combo tested in vitro and to demonstrate that they could have a certain clinical significance.

Finally, the authors used the term “perspectives” in the title, but they omitted to develop the perspectives of their study. If they consider that an in vitro assay could be sufficient to determine the chemotherapy to select for treating patient, what is the strategy they propose ? What could be the cut-off of in vitro sensitivity for defining the treatment ?

No comment

Author Response

Dear Peer Reviewer,

We are very grateful for your careful reading of our manuscript, its analysis and useful comments. Please find our responses on your comments.

Comment 1: The authors underlined the fact that development of personalized methods predicting the tumor’s response are needed for optimizing treatment by chemotherapy. However, in Introduction, they completely omitted to mention all genomic and transcriptomic investigations. Now NGS and kits for detecting mutations are usually used for profiling patient. As an example, for some cancers, detection of KRas mutations is now used for defining treatment of patients. These methods will be far more standardizable than in vitro tumor cell culture. Introduction should be enriched by presenting the other approaches.

Response 1: Thank you very much for the comment, the Introduction has been rewritten and we have added the following information:

Currently, there are no clinically used molecular markers that predict response to therapy and survival in STS. The data obtained on some STS cell lines [9] are not in full agreement with the results of tumor analysis of STS patients [10]. Although the mutation rate is low in STS [11], a number of studies have revealed a relationship between the development of resistance to doxorubicin and the presence of mutations in the apoptosis activation path-way [12]. It was found that in ChT-resistant SMT samples, the expression of genes in-volved in signaling pathways associated with cell adhesion, migration, proliferation, cy-totaxis and phagocytosis, as well as signaling cascades PI3K-Akt, NF-ĸB, MAPK and Jak –STAT is activated, and then C1QC, C3AR1, C7, CFI, and SERPINE1 were selected as po-tential ChT resistance and survival biomarkers [13]. Lindner et al. showed that high ex-pression of TOP2A was associated with low overall survival, while high expression of SIRT1 was associated with high overall survival [14], according to the authors, the pre-sented markers may be of interest for predicting patient survival with SMT.

An alternative approach of experimental ChT-resistance testing by the Cell Sensitivity assay (CSA) was proposed by Kurbacher at al. for ovarian cancer [15]. Using CSA, a num-ber of studies of tumors of different histogenesis have been published, and a high prog-nostic value of detecting resistance to chemotherapy has been demonstrated for different cancers [16]. Volm and Efferth paid more attention to in vitro testing of the ChT-resistance than to ChT-sensitivity as in several clinical trials, ChT-resistance was predictable with more than 90% accuracy, while drug sensitivity was detected with significantly less ac-curacy (~60%). Consequently, we mainly directed our study to the ChT-resistance predic-tion to develop in future the approach for more safe chemotherapy by preventing useless and harmful treatment of STS patients, and extended the abbreviation CSA, proposed by Kurbacher at al.  to CSRA (Cell Sensitivity/Resistance Assay).

CSA was used for STS only in a few studies. M. Lehnhardt et al. studied the correla-tion between the CSA data to ChT drugs, tumors histotype, and malignancy grade using tumor samples of 50 STS patients [17]. In later years, two clinical cases using the studied approach were published detailing the selection of neoadjuvant ChT regimens in patients with synovial sarcoma [18]] and undifferentiated pleomorphic sarcoma [19].  In our pre-vious studies, using CSRA test, we showed that mutations in apoptosis signaling path-ways are associated with resistance to doxorubicin [12], and it was also shown that the expression of ABC transporters in adult SMT does not play a significant role in ChT-resistance [10]. Oncomedics have proposed a similar to CSA method for ChT-resistance experimental testing in colon cancer [20] and ovarian cancer [21]. There is a number of differences between Oncomedics and CSA, including different protocols of tumor samples treatment, seeding and culturing conditions. Currently we perform the study to compare these methods

Comment 2:. It is also important to notice that the approach the authors propose is already commercialized by Oncomedics. That could be mentioned and discussed if the techniques proposed is not strictly identical.

Response 2: Information concerning the method used by Oncomedics has been added to the introduction, however, we also pointed out that Oncomedics protocol has been applied only for colorectal cancer and ovarian cancer and previously it has not been used for SMT.

Comment 3:  Authors completely omitted to discuss all the limitations of the in vitro tumor cells. Many CROs try to transfer primary tumor cell for establishing PDX or for performing in vitro experiments. They all observed that in vitro adaptation is not uniform: some adaptation are easy but with some other tumor, cells do not grow, and it is difficult to not consider a case-by-case situation. May be that this predictive approach is more appropriate for certain tumors but this question was even never evoked. These points have to be discussed and some in vitro curves have to be presented in a novel figure (or at least as supplementary data).

Response 3: The method used in this study involves a test to determine the sensitivity/resistance to ChT immediately after the isolation of cells from the tumor. At the same time, the exposure with drugs lasts 7 days. For patients immediately after and after 7 days of culturing, a cytological analysis was performed to determine the percentage of tumor cells (more than 95%), it was shown that their number increases, in addition, the cells obtained in parallel are still cultivated for some time, for our studies (at least 3 passages). Thus, by the time the test is completed, the cells are still able to divide. In addition, the method proposed by Oncomedics suggests cells culturing for a week before testing. We are currently conducting a study that will compare the approaches of Oncomedics and CSRA.

Comment 4: The question of the in vitro adaptation is also important because stem cell-like cells usually do not grow in vitro … but they have a major role in clinical resistance. This bias is completely omitted.

Response 4: Thank you very much for your question. Yes, indeed, the acquisition of a stem phenotype by cancer cells plays a role in the development of resistance to chemotherapy associated with the development of relapses. CSA minimizes the influence of the factors mediated by organism and the interaction of cancer cells with the microenvironment, as well as the acquisition of resistant phenotypes by cancer cells, which includes stem and hypoxic phenotypes, which may impose some limitations on the prognostic value of CSRA predicted sensitivity. Thus, you would be exactly right, if we directed our study for determination of sensitivity, we could miss resistance caused by the acquisition of a stem phenotype and have a wrong sensitivity prediction. However, Gong at al demonstrated that in ChT-resistant SMT samples, the activity of signaling pathways associated with cell adhesion, migration, proliferation, cytotaxis, and phagocytosis most often changes. In addition, such signal cascades as PI3K-Akt, NF-ĸB, MAPK and Jak-STAT played in the development of resistance in SMT cells [10.1016/j.ejphar.2020.173121]. Thus, the acquisition of a stem phenotype by cancer cells and the microenvironment influence do not represent the only mechanisms of ChT resistance. The main goal of our study is to reveal cancer cells resistant to the drug in vitro to predict inefficacy of possible ChT. It is in perfect agreement with the conclusion of Volm and Efferth [10.3389/fonc.2015.00282] concerning ChT resistance revealing by CSA, which predictive significance is about 90 %.

However, in our future experiments, we plan to analyze the possibility of using 3D culturing methods, including enrichment of isolated cultures with cells with a stem phenotype and comparing these methods with the clinical response of patients.

Following your comment, we have added corresponding information to the Discussion.

Comment 5: Sensitivity of drugs in vitro are mentioned but their mode of action not. It is well known that depending on the mode of action, optimal drug concentration and incubation time are not the same. Therefore, the meaning of global and statistical analysis without any consideration of this point is questionable.

Response 5: We have added the information and the following references:

  • Whitehouse PA, Knight LA, Di Nicolantonio F, Mercer SJ, Sharma S, Cree IA; Portsmouth Colorectal Cancer Multidisciplinary Team. Heterogeneity of chemosensitivity of colorectal adenocarcinoma determined by a modified ex vivo ATP-tumor chemosensitivity assay (ATP-TCA). Anticancer Drugs. 2003 Jun;14(5):369-75. doi: 10.1097/00001813-200306000-00008. PMID: 12782944.
  • Chen Z, Zhang S, Ma S, Li C, Xu C, Shen Y, Zhao J, Miao L. Evaluation of the in vitro Chemosensitivity and Correlation with Clinical Outcomes in Lung Cancer using the ATP-TCA. Anticancer Agents Med Chem. 2018;18(1):139-145. doi: 10.2174/1871520617666170419123713.
  • Cree IA, Neale MH, Myatt NE, de Takats PG, Hall P, Grant J, Kurbacher CM, Reinhold U, Neuber K, MacKie RM, Chana J, Weaver PC, Khoury GG, Sartori C, Andreotti PE. Heterogeneity of chemosensitivity of metastatic cutaneous melanoma. Anticancer Drugs. 1999 Jun;10(5):437-44. doi: 10.1097/00001813-199906000-00002.

Comment 6: It is extremely difficult to compare combination of drugs and in clinic and in vitro. Chronology and doses of clinical treatment are far different than those of coincubation in vitro, especially with drugs that induce apoptosis after a cell cycle blockage. It essential to well explain the rational for establishing the combo tested in vitro and to demonstrate that they could have a certain clinical significance.

Response 6: We followed the protocol details concerning drug combinations described in the previous studies:

1)         Ling ZQ, Qi CJ, Lu XX, Qian LJ, Gu LH, Zheng ZG, Zhao Q, Wang S, Fang XH, Yang ZX, Yin J, Mao WM. Heterogeneity of chemosensitivity in esophageal cancer using ATP-tumor chemosensitivity assay. Acta Pharmacol Sin. 2012 Mar;33(3):401-6. doi: 10.1038/aps.2011.195.

2)         Neubauer H, Stefanova M, Solomayer E, Meisner C, Zwirner M, Wallwiener D, Fehm T. Predicting resistance to platinum-containing chemotherapy with the ATP tumor chemosensitivity assay in primary ovarian cancer. Anticancer Res. 2008 Mar-Apr;28(2A):949-55.

In connection with your comment, we also have made changes to the discussion.

Comment 7: Finally, the authors used the term “perspectives” in the title, but they omitted to develop the perspectives of their study. If they consider that an in vitro assay could be sufficient to determine the chemotherapy to select for treating patient, what is the strategy they propose? What could be the cut-off of in vitro sensitivity for defining the treatment?

Response 7:. Thank you for this comment. Following it, we have added the corresponding information to the discussion:

The in vitro CSRA, used in our study for STS, can become an effective tool for prediction of tumor ChT resistance to anticancer drugs and their combinations, followed by correction of the ongoing treatment by withdrawal of inefficient drug from the combination. It is in agreement with the case reports published by Cao et al. [31815036] and then Florou and Wilky [210.1007/s11864-021-00858-9] demonstrating that when conducting chemotherapy for soft tissue angiosarcomas, the question was which combination of drugs are most likely to be efficacious to be chosen for the first line: Dox+IFO or Dots+Gem, and CSRA approach may be useful in providing an informed choice between these combinations. In addition, CRSA performed simultaneously with molecular biological studies on the obtained primary cell cultures of STS patients can be used both to search for prognostic biomarkers and to analyze the development of resistance mechanisms in SMT cells. We have already shown the effectiveness of an approach to studying the mechanisms of development of multidrug resistance in soft tissue sarcomas based on the use of primary cultures of tumor cells [10.3390/cancers14071796, 10.3390/ijms23063183]. Thus, CSRA application in STS study and management will contribute to a more rapid progress of knowledge in this field.

Round 2

Reviewer 3 Report

In this version many points are clarified and scientific approach of the authors is more readable.

Still few remarks:

Figure 1 and 2 could be enlarged to be easier to read

Table 2 remains unclear and should be comment in more details.

There are so many abbreviations that a list should be added.

Some typos to be check in this version.

The question of the combination remains important to comment. Mat & Met suggest that the 2 drugs are co-incubated, that is not really the case in clinic. In vitro experiments to compare co-incubation and sequential treatment have usually showed different results and even opposite for some of them. This results from the mode of action of each agent and their kinetics of action. The choice to co-incubate the 2 drugs should not result only from the decision to reproduce a method already published without any reflexion regarding this point. Co-incubation can be used but with reasonable arguments.

No major remark

Author Response

Dear Peer Reviewer,

We are very grateful for your careful reading of our manuscript, its analysis and useful comments. Please find our responses on your comments.

Comment 1: Figure 1 and 2 could be enlarged to be easier to read.

Response 1: The Figures 1 and 2 have been enlarged

Comment 2: Table 2 remains unclear and should be comment in more details.

Response 2: Additional details have been inserted into table 2.

Comment 3:  There are so many abbreviations that a list should be added.

Response 3: The abbreviation list has been added.

Comment 4: Some typos to be check in this version.

Response 4: The text of the article has been checked and typos have been corrected.

Comment 5: The question of the combination remains important to comment. Mat & Met suggest that the 2 drugs are co-incubated, that is not really the case in clinic. In vitro experiments to compare co-incubation and sequential treatment have usually showed different results and even opposite for some of them. This results from the mode of action of each agent and their kinetics of action. The choice to co-incubate the 2 drugs should not result only from the decision to reproduce a method already published without any reflexion regarding this point. Co-incubation can be used but with reasonable arguments.

Response 5: In our work we applied one of the most published approach for predictive chemoresistance testing with the high predictive value, which was demonstrated in the studies of Ian A. Cree, a very famous British oncologist (he has published more than 50 papers devoted to this test application among his 423 publications, and these publications are highly cited in other studies). To reproduce this methodology, we followed the most used published protocol. However, you are exactly right concerning reasonable arguments and justification of our CSRA protocol in respect of STS. In STS chemotherapy the most widely used protocol includes simultaneous treatment by Doxorubicin and Ifosfamid at the first day and then two more days Ifosfamid treatment. Thus our approach is concordant with the protocol used in chemotherapy of STS patients.

The protocol used in the N.N.Blokhin NMRCO:

Drugs used in the protocol

Day

1

2

3

Doxorubicin (alone)

75 mg/m2

Ifosfamide (alone)

3 g/m2

3 g/m2

3 g/ m2

Doxorubicin

+

Ifosfamide

75 mg/m2

+

3 g/m2

g/m2

3 g/ m2

Judson I, Verweij J, Gelderblom H, Hartmann JT, Schöffski P, Blay JY, Kerst JM, Sufliarsky J, Whelan J, Hohenberger P, Krarup-Hansen A, Alcindor T, Marreaud S, Litière S, Hermans C, Fisher C, Hogendoorn PC, dei Tos AP, van der Graaf WT; European Organisation and Treatment of Cancer Soft Tissue and Bone Sarcoma Group. Doxorubicin alone versus intensified doxorubicin plus ifosfamide for first-line treatment of advanced or metastatic soft-tissue sarcoma: a randomised controlled phase 3 trial. Lancet Oncol. 2014 Apr;15(4):415-23. doi: 10.1016/S1470-2045(14)70063-4. Epub 2014 Mar 5. PMID: 24618336.

Maki RG, Wathen JK, Patel SR, Priebat DA, Okuno SH, Samuels B, Fanucchi M, Harmon DC, Schuetze SM, Reinke D, Thall PF, Benjamin RS, Baker LH, Hensley ML. Randomized phase II study of gemcitabine and docetaxel compared with gemcitabine alone in patients with metastatic soft tissue sarcomas: results of sarcoma alliance for research through collaboration study 002 [corrected]. J Clin Oncol. 2007 Jul 1;25(19):2755-63. doi: 10.1200/JCO.2006.10.4117. Erratum in: J Clin Oncol. 2007 Aug 20;25(24):3790. PMID: 17602081.

We added the corresponding information in the text and appreciate your comment very much.